# Clinical Impact of Mediterranean Diet Adherence before and after Bariatric Surgery: A Narrative Review

**DOI:** 10.3390/nu14020393

**Published:** 2022-01-17

**Authors:** Isabella Gastaldo, Rosa Casas, Violeta Moizé

**Affiliations:** 1Endocrinology Department, Hospital Clínic de Barcelona, c/Villarroel, 170, 08036 Barcelona, Spain; isabellapgastaldo@gmail.com; 2Department of Internal Medicine, Hospital Clinic de Barcelona, c/Villarroel, 170, 08036 Barcelona, Spain; rcasas1@clinic.cat; 3Institut d’Investigacions Biomèdiques August Pi Sunyer (IDIBAPS), c/Rosellon, 149, 153, 08036 Barcelona, Spain; 4Centro de Investigación Biomédica en Red de la Obesidad y la Nutrición (CIBEROBN), Instituto de Salud Carlos III, 28029 Madrid, Spain; 5Centro de Investigación Biomédica en Red de Diabetes y Enfermedades Metabólicas Asociadas (CIBERDEM), Instituto de Salud Carlos III, 28029 Madrid, Spain

**Keywords:** Mediterranean diet, obesity, bariatric surgery, weight loss

## Abstract

The population suffering from obesity is rapidly increasing all over the world. Bariatric surgery has shown to be the treatment of choice in patients with severe obesity. A Mediterranean diet has long been acknowledged to be one of the healthiest dietary patterns associated with a lower incidence of many chronic diseases. The aim of the present narrative review is to summarize the existing research on the clinical impact of a Mediterranean diet before and after bariatric surgery, focusing on its effects on weight loss and improvement in comorbidities. Although the current knowledge is limited, this information could add value and emphasize the importance of adopting a Mediterranean diet before and after bariatric surgery.

## 1. Introduction

In an age where people’s days are increasingly busy, finding the time to incorporate a healthy lifestyle is paramount. When the focus is prevention, healthy eating habits and regular physical activity are the key factors that contribute to the health of populations, especially with regards to chronic disease [1,2]. Over the past decades, there has been a real explosion in the worldwide prevalence of obesity [3], with a significant increase in the amount of research being published, with over 390,000 papers on this topic. Unfortunately, despite the numerous health programs, campaigns and wide dissemination of information focusing on prevention, things are not improving. The World Health Organization (WHO) predicts that the number of obese people will continue to increase steadily over the next years [4]. It is estimated that the number of overweight and obese adults by 2030 to reach 2.16 billion overweight and 1.12 billion obese individuals [5]. That is, obesity has become the 21st century’s epidemic, with more than 100 million euros spent on health care costs over a lifetime for 1000 people with morbid obesity [6].

When conventional treatments for obesity result insufficient, bariatric surgery (BS) has been recognized as an effective alternative for weight loss in subjects with severe obesity and has become one of the most common surgical procedures performed worldwide, according to recently published data [7]. Between all the different surgical techniques, laparoscopic Roux-en-Y gastric bypass (LRYGB) and laparoscopic sleeve gastrectomy (LSG) are the most widely used and have shown similar long-term results on weight loss and resolution of comorbidities [8]. In addition, something that is often overlooked, is weight loss before surgery, also known as preoperative weight loss. It is important due a number of factors, including control of fatty enlargement of the left lobe of the liver, which acts to obscure the operating field, and the mitigation of the risk of bleeding during the operation, therefore reducing postoperative complications and mortality [9]. Even with surgery, however, sometimes the weight loss is less than expected, or comorbidities do not improve [10,11,12]. This has led researchers to investigate potential strategies to overcome such issues.

It has long been demonstrated that adherence to a Mediterranean diet (MD) is associated with longevity, and a lower risk of suffering from obesity [13,14,15]. It is one of the most recognized healthy dietary patterns, characterized by a high intake of fruits and vegetables, legumes, nuts, whole grains, extra virgin olive oil, a moderate consumption of fish and poultry, with a low intake of red and processed meats, sugar-sweetened beverages, and processed foods (Figure 1). Unfortunately, to the best of our knowledge, very few studies have been performed with regards to the effects of a Mediterranean diet on bariatric surgery outcomes (see Table 1 for a summary).

Given the established benefits of adhering to a Mediterranean diet, the aim of this review was to further contribute to this growing area of research by summarizing all of the current evidence available on the effects of a Mediterranean diet in obese patients before or after bariatric surgery, and its impact on surgery outcomes such as weight loss. This may serve as proof to encourage a certain dietary pattern after bariatric surgery.

## 2. Materials and Methods

### 2.1. Literature Search

A literature search of PubMed, Cochrane, and Google Scholar was conducted until October 2021, without restriction of date. Searches were performed using key search terms related to bariatric surgery and Mediterranean diet. These included the following terms: “bariatric surgery”, “gastric bypass”, “Roux-en-Y”, “sleeve gastrectomy”, “adjustable gastric banding”, “Mediterranean”, “Mediterranean dietary pattern”, and “Mediterranean diet”. With respect to article types, only studies conducted in human subjects were reviewed and a possible limitation was the publication language, as only English and Spanish articles were included.

### 2.2. Study Eligibility and Selection

Studies were considered eligible for inclusion in the present narrative review if they assessed adherence to a Mediterranean diet before or after bariatric surgery independent of whether they evaluated its impact on weight loss. The initial screening of articles was done by reviewing titles and abstracts to determine eligibility. Studies that did not satisfy the inclusion criteria were excluded (Figure 2). Due to the limited amount of research on this topic, all articles covering bariatric surgery and Mediterranean diet were included.

## 3. Results

### 3.1. Mediterranean Diet Adherence before Bariatric Surgery

In order to evaluate the effects of a Mediterranean diet in patients undergoing LSG, a group of researchers conducted an 8-week study with 37 obese male patients [17]. Up to this point few researchers had begun exploring the effects of preoperative eating patterns on short-term and mid-term weight loss [21], and there had been no studies investigating a preoperative Mediterranean diet. Patient characteristics for this descriptive prospective cohort were obese men, with a body mass index (BMI) ≥ 40 kg/m^2^, aged 25–65 years, married, living at home with their parents, or living with a partner in a stable relationship for at least 1 year.

An important observation to make, is that this was a Mediterranean-protein-enriched diet (MPED), having a macronutrient distribution of 30% protein, 25% fat, and 45% carbohydrate, as opposed to a more traditional Mediterranean diet, which has a distribution of 15% protein, 35% fat, and 50% carbohydrate [22]. Each participant followed a diet for 8 weeks and was asked to complete a series of assessments. The way in which researchers made sure that all participants consumed a similar diet was by developing four meal plans, one for every two weeks, which were composed of pre-defined fruits, vegetables, pasta, milk products, herbs and spices, as well as meat and fish. 

Dietary assessments consisted of 3-day food records completed for 3 consecutive days, a daily food diary, and 72-h recalls conducted by a trained dietician. To ensure compliance, urine samples were collected biweekly and used with urinary ketone reagent strips to determine a total ketone score. Qualitative methods were also used to measure diet acceptability using a five-point Likert scale. Lastly, weight, visceral fat, body composition, liver size, and biochemical and metabolic patterns were measured before and after the 8-week intervention. 

Before the intervention, all participants showed an excessive intake of saturated fats, mostly in the form of red meat, a high intake of cholesterol-rich foods, such as eggs and cheese, and an extreme consumption of sweet-carbonated beverages. They were also observed to have a low consumption of nutrient rich foods, such as fruits, vegetables, legumes, whole grains, nuts, olive oil and fish.

After 8-weeks of following a Mediterranean diet, the authors observed significant reductions (*p* = 0.01) in weight (−16.7%), visceral fat (−27.4%), left and right liver lobe sizes (−29.1% and 25.2%, respectively), and total fat mass (FM) loss (14.1%), without a significant reduction (*p* = 0.0960) in fat-free mass (FFM). Clinical parameters such as liver enzymes, and lipid profiles also showed improvements. The total ketone score was found to be highly correlated with the total percentage weight reduction at the end of the intervention. This study demonstrated that a Mediterranean diet might be beneficial for bariatric surgery candidates looking to lose weight while maintaining FFM. 

### 3.2. Mediterranean Diet Adherence after Bariatric Surgery

When it comes to the effects of adhering to a Mediterranean diet both in the pre- and post-operative stage, in 2014 a group of Spanish researchers published a prospective observational study which evaluated the adherence of a Mediterranean diet in morbidly obese patients before and after undergoing sleeve gastrectomy [18]. A total of 50 patients, 44 women and 6 men, with a mean BMI of 50.4 kg/m^2^ were included. Of these, 50% presented with dyslipidemia, 30% with hypertension, 28% with type 2 diabetes (T2D), 24% with osteoarthritis, and 16% with sleep apnea syndrome. The overall aim of the study was to analyze the association between Mediterranean diet adherence and postoperative weight loss, resolution of comorbidities, as well as cardiovascular risk factors.

The preoperative tests performed on each patient included abdominal ultrasound, upper gastrointestinal endoscopy, spirometry, and blood analysis, which also included nutritional parameters. Patients were prescribed a Mediterranean diet of 1200 Kcal preoperatively and were asked to follow itfor 2 months to achieve a weight loss of at least 10%—a requirement to proceed with the surgery. All patients completed the KIDMED test before the operation and 1 year after the intervention. The KIDMED test measures adherence to the Mediterranean diet, analyzing Mediterranean diet patterns such as daily consumption of fruits and vegetables, weekly intake of fish and legumes, as well as patterns contrary to the Mediterranean diet, such as the frequent consumption of meat, processed foods, and sweets. The test is composed of 16 questions, answered by a simple “yes” or “no”. A total score between 0–3 reflects poor adherence, between 4–7 moderate, and between 8–12 good adherence.

One year after surgery, patients had an average of 81.3% weight loss, a significant reduction in blood glucose levels, with a *p* value of 0.003 (mean decrease of 33.2 mg/dL), decreased levels of glycated hemoglobin, (*p* = 0.08), decreased triglycerides values (*p* = 0.001), and an insignificant decrease in low density lipoprotein (LDL) cholesterol with a significant increase (*p* < 0.001) in high density lipoprotein (HDL) cholesterol. Additionally, while the preoperative mean Mediterranean diet adherence score was 4.5, the postoperative mean score was 7, with a mean increase of 2.5 points (*p* < 0.001). This means that before surgery, only 6% of patients had a good Mediterranean diet adherence, a number that changed to 40% after 1 year (*p* = 0.02). 

An inverse correlation was observed between KIDMED scores and weight loss, meaning patients who showed better adherence to the Mediterranean diet presented greater weight loss. The same was true for total and LDL-cholesterol, which decreased as KIDMED scores increased. On the same note, a direct correlation was established between KIDMED scores and HDL-cholesterol levels, where patients with higher scores had increased HDL values. Lastly, although not statistically significant, all comorbidities such as T2D, arterial hypertension and dyslipidemia were improved, with patients with better adherence showing greater improvements. This was the first study to evaluate adherence to a Mediterranean diet in patients with severe obesity both before and after bariatric surgery, and to analyze its effects on weight loss and comorbidities.

The second, and only other study which evaluated the effects of a Mediterranean diet after bariatric surgery was published in 2020 [19]. This was another prospective observational study aimed at assessing how changes in dietary food pattern and physical activity affect changes in weight, BMI, quality of life, and food tolerance after surgery. A total of 78 participants, 59 women and 19 men, aged between 18 and 66 years, with a BMI ≥ 35 kg/m2 were included. After surgery, researchers provided leaflets with general dietary recommendations on a healthy diet. Although the study measured both Mediterranean diet adherence and physical activity, for the purposes of this review, only outcomes related to a Mediterranean diet were explored.

Adherence to a Mediterranean diet, as well as weight, quality of life, and food tolerance were measured at baseline, 3 weeks before surgery, and at 3, 6, 9, and 12 months after. Mediterranean diet adherence was estimated using a validated 14-point method called Mediterranean Diet Adherence Screener (MEDAS), composed of 14 questions, which scores one if the answer adheres to a Mediterranean diet, and zero if it does not. Therefore, final scores range from 0–14 and are subdivided into: very low adherence (0–4), low adherence (5–7), medium adherence (8–11), and high adherence (12–14). Scores were dichotomized into an increase, a decrease, or maintenance of adherence to a Mediterranean diet.

At baseline, 59.2% of patients had a low or very low adherence to the Mediterranean diet. After a 12-month follow up, participants who increased adherence had a significantly higher weight loss (−48.7 kg vs. 43.9 kg), and percentage of excess weight loss (−71.2% vs. −63.1%) than those who decreased or maintained adherence (*p* = 0.036). This study also demonstrated that individuals who adhere to a Mediterranean diet after undergoing bariatric surgery present with greater weight loss than those who do not. However, a possible limitation of this study is that it did not use biomarkers to assess adherence, something that the above studies did.

### 3.3. Long Term Adherence to a Mediterranean Diet

There has been only one paper in the literature, published in 2020, which explored long term dietary habits of a cohort of patients who had undergone sleeve gastrectomy with a follow-up of at least 4 years by comparing it to the Italian Mediterranean diet (IMD) recommendations [20]. The main difference between a traditional MD and the IMD is that at the bottom of the pyramid, instead of having any type of cereal, such as bread, pasta, rice, or couscous, only cereals which have a low glycemic index and are rich in fiber are included. These include whole meal wheat and pasta, sourdough bread, whole grains, and brown rice [23]. Relatedly, at the top of the Italian pyramid are cereal foods which are poor in fiber such as white bread, white rice, couscous, white pasta, and potatoes. Although the classic Mediterranean diet encourages whole grains, it is not a crucial component of its structure. 

In this study, a total of 74 patients, of which 78.4% were females, with a mean BMI of 45.7 kg/m^2^ were included. The aim of the study was to investigate dietary habits, weight, BMI, evolution of comorbidities, and micronutrient status and supplementation. To do so, the authors assessed dietary habits by a 7-day food diary (guided by a dietician), body weight, obesity related comorbidities, as well as micronutrient status and use of micronutrient supplements. Blood tests were used to explore micronutrient deficiencies. Lastly, an important parameter measured was weight regain, and such was defined as gaining at least 15% of the initially lost weight after surgery.

At follow-up, the results were surprising. An astonishing 74.3% of patients reported to consume at least one sweetened-carbonated beverage each day. Only 40.5% of patients followed the IMD recommendations for fruits, 35.1% for vegetables, and 40.5% for whole grains. To make matters worse, only 35.1% of patients reported eating the weekly recommendations for legumes, 13.5% using herbs, spices, garlic, or onions, and a merely 5.4% stated to consume the daily amounts of nuts, seeds, or olives. This dietary pattern demonstrates an inadequate adherence to a Mediterranean diet in the most fundamental ways. 

Before surgery, mean body weight and BMI were 141.7 kg and 45.7 kg/m^2^, respectively. At the time of follow-up, at least 4 years after surgery, mean body weight and BMI had both decreased significantly to 76.7 kg and 25.7 kg/m^2^, respectively. Pre-operative type 2 diabetes was present in 20.3% of patients, of which 73.3% reported either a reduction or termination of antidiabetic drugs. Moreover, 23% of patients had hypertension, with 64.7% of them reporting either a reduction or termination of antihypertensive drugs. Finally, 54% of patients had obstructive sleep apnea, a number which dropped to zero at the time of follow-up. As can be seen here, majority of patients showed an improvement in the evolution of comorbidities after bariatric surgery, as can be expected after such intervention [24,25,26,27].

However, it is essential to note that weight regain was found in 37.8% of patients. Of these, 32% showed an inadequate adherence to an IMD by not following recommendations for fruits, vegetables, white rice, pasta, bread, and potatoes. Additionally, only 18.9% of patients were taking micronutrient supplements and several deficiencies were found, including vitamin D, B12, folate, and iron. This study further demonstrates that individuals who adhere to a Mediterranean diet show greater weight loss than those who do not, or more importantly, they avoid regaining the weight that had been lost after surgery.

## 4. Discussion

With this review, we aimed to summarize the current evidence on the links between Mediterranean diet adherence and bariatric surgery outcomes. Overall, the combined papers included suggest that following a Mediterranean diet before or after bariatric surgery can influence weight loss, and resolution of comorbidities. Our effort was directed by the limited number of studies which have been published thus far but are indicative of strong evidence in favor of a Mediterranean diet as all studies are in agreement with each other.

The most important finding was that all studies showed that patients who had better adherence to a Mediterranean diet lost more weight. This outcome is understandable since it has long been suggested that a Mediterranean diet is a suitable dietary pattern for many purposes, including weight loss and metabolic health [28,29,30]. It is important to also point out that the PREDIMED study, a randomized control trial (RCT) with 4.8-years of follow-up, that included 7447 participants at high risk for cardiovascular disease (CVD), showed that participants with high adherence to a MD and a modest weight loss significantly reduced their risk of CVD and T2D [31,32]. However, since weight regain is a major concern following bariatric surgery [33,34], having the knowledge that perhaps patients should be following a Mediterranean diet, not only pre-surgery but also post-surgery, adds tremendous value to patient care. It is evident that bariatric surgery is a great tool for weight loss and improvement of comorbidities, as demonstrated in the above papers, but following a Mediterranean diet also seems to be of extreme significance.

A common theme emerged between the papers. At baseline, the great majority of patients showed a poor adherence to a Mediterranean diet, characterized notably by a low consumption of fruits and vegetables, with a high consumption of sweetened carbonated beverages. This is a concerning issue, as fruits and vegetables provide a great source of nutrients [35,36] and have even been demonstrated to correlate with weight and fat loss [37]. On the same note, a recent systematic review and meta-analysis demonstrated that there is a significant association between both sugar and artificially sweetened soda consumption and obesity, suggesting that soda consumption should be kept to a minimum [38]. The low consumption of fruits and vegetables, together with the high consumption of sweetened beverages show inadequate dietary patterns for populations trying to maintain a healthy lifestyle. In the same context, it is important to notice that food addictions and binge eating are common issues in overweight and obese patients and should be addressed before they undergo surgery [39].

The Spanish Federation of Nutrition, Food and Dietetics Associations (FESNAD) and the Spanish Association for the Study of Obesity (SEEDO) have come together to establish nutritional recommendations for the prevention and treatment of overweight and obesity in adults, based on a systematic review of 15 years of published data [40]. Evidence regarding feeding patterns and body weight point to a potential role for a MD in the prevention of overweight and obesity, as well as decrease in abdominal circumference. They have also suggested that vegetarian diets may lead to less weight gain over time, and this might be explained by a high intake of fruit and vegetables, which have been associated with smaller weight increase in adults in the long term. The high intake of whole grains was also associated with a lower BMI, whereas the frequent intake of sugar-sweetened beverages was associated with a higher BMI. Therefore, including pre- and post-operative dietary patterns when studying the effectiveness of BS, would result in more accurate findings and would result in better care [41], while more frequent follow-up visits after BS may help patients reduce weight regain in long-term management [42].

The strength of this review consists in its attempt to present and summarize all evidence regarding the effects of a Mediterranean diet on bariatric surgery outcomes. We have demonstrated that weight loss occurs as a result of adhering to a Mediterranean diet even before patients undergo bariatric surgery, and that weight loss is not dependent on it. Secondly, even though this was a bibliographic, non-systematic review, we believe that it can be used to demonstrate the importance of adhering to a Mediterranean diet, which favors weight loss and improvement of comorbidities. Finally, the entire population studied in this review was from a Mediterranean region, more specifically Spain and Italy. This addresses the concern that people might not adhere due to various barriers and accessibility, since such dietary pattern is culturally promoted in these regions.

The main limitation of this review is its narrative and not systematic nature, due to the difficulty in finding homogeneous data on this topic, particularly on weight loss prior to surgery. Most of the studies considered were retrospective, not randomized, and heavily influenced by health insurance directives. Different panels of clinical tests required by different teams prior to surgery have been another limitation for a systematic evaluation. On the same note, dietary assessment methods varied between the papers presented and this may have influenced the results and conclusions of the studies. For example, the assessment of MD adherence varied from 3-day questionnaires, KIDMED, MEDAS, and food dietary records, demonstrating an inconsistent methodology. Similarly, the heterogeneity in methodology and populations used might have contributed to the outcomes presented. Lastly, remission of comorbidities could not have been attributed solely to a Mediterranean diet, as even patients with poor adherence showed remission for some conditions, such as hypertension. The most likely explanation is that improvement of comorbidities happened because of weight loss. Nonetheless, what has become apparent is that a Mediterranean diet is undoubtedly a good dietary pattern for weight loss before and/or after bariatric surgery.

## 5. Conclusions

Overall, the evidence suggests that adherence to a Mediterranean may be directly connected with weight loss before or after bariatric surgery. Patients which show greater adherence lose more weight than those who show worse. Despite the limited number of studies, the effects of a Mediterranean diet on obese patients are promising. Still, with the decline of people following a Mediterranean diet, even in Mediterranean countries, it is crucial that we continue to support and recommend such dietary pattern. It is important to observe that weight loss and comorbidity evolution after bariatric surgery are multifactorial in nature as many factors can influence these outcomes (e.g., physical activity, initial BMI, socioeconomic status, psychosocial and cognitive factors, and social support), and the observational studies described above could not have ruled out these confounders. Therefore, further research is needed to establish, with certainty, if a Mediterranean diet can be used to ensure resolution of comorbidities, good micronutrient status, and body weight maintenance at long-term.

## Figures and Tables

**Figure 1 nutrients-14-00393-f001:**
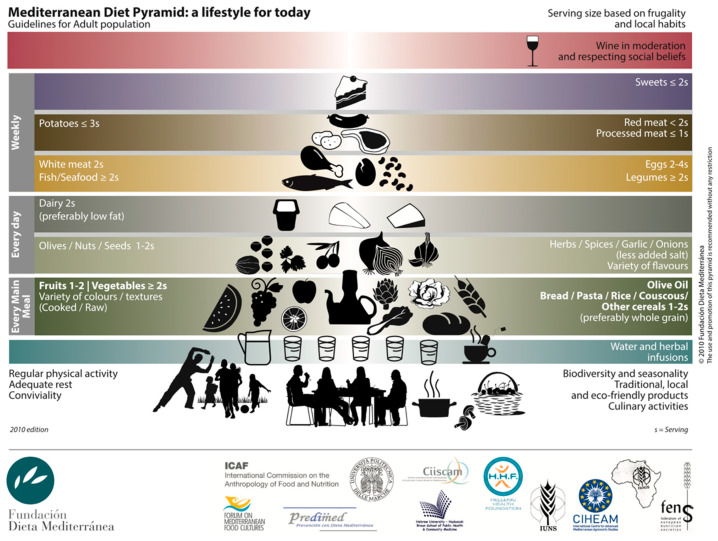
Composition of a Mediterranean Diet. Adapted from [16].

**Figure 2 nutrients-14-00393-f002:**
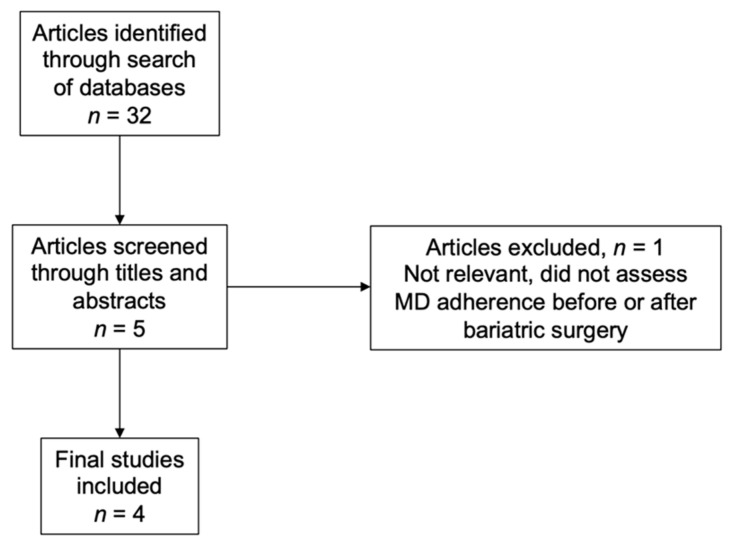
Selection Process for Articles.

**Table 1 nutrients-14-00393-t001:** Mediterranean Diet Adherence and Bariatric Surgery Outcome.

Reference	Year	Study Design	Number of Participants	Follow-Up (Years)	Main Findings
Schiavo et al. [17]	2015	descriptive prospective cohort	37	--	Participants who adhered to a MD ^1^ showed significant decreases in weight, liver size, visceral fat, and fat mass before undergoing BS ^2^.
Ruiz-Tovar et al. [18]	2014	prospective observational study	50	1	Better adherence to a MD resulted in greater weight loss, and improved lipid profiles after BS.
Contreras et al. [19]	2020	prospective observational study	78	1	Increased adherence to a MD resulted in greater weight loss after BS.
Schiavo et al. [20]	2020	prospective observational study	74	4	Patients who underwent BS were found to have an inadequate MD adherence after at least 4 years of intervention, with weight regain in 37.8% of patients.

^1^ Mediterranean diet, ^2^ Bariatric surgery

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
