# Peer review of "Clinical Impact of Mediterranean Diet Adherence before and after Bariatric Surgery: A Narrative Review"

_nutrients, 2022, doi:10.3390/nu14020393_

Round 1

Reviewer 1 Report

This is a narrative review is to summarize the existing research on the clinical impact of a Mediterranean diet before and after bariatric surgery,
focusing on its effects on weight loss and improvement of comorbidities.
After literature research, four studies were included for analysis.
The author concluded that adherence to a Mediterranean diet is directly correlated with weight loss before or after bariatric surgery. 
Patients which show greater adherence lose more weight than those who show worse.

However, the author did not provide detailed content of each included study.
The author used only a plan description regarding the study result and
we also cannot see any convincing statistical analysis.

Therefore, the conclusion is not objective and the author should provide more details of each included study and statistical analysis to support the conclusion.

Reviewer 2 Report

The manuscript presented a narrative review, summarizing existing research on the clinical impact of a Mediterranean diet before and after bariatric surgery on weight loss and improvement of comorbidities.  

I have a few comments. First, the manuscript situated Mediterranean Diet (MD) around bariatric surgery, however a balanced discussion of the role of weight loss diets before and after bariatric surgery appeared to be missing. For example, one of the main purposes of preoperative weight loss is to control the fatty enlargement of the left lobe of the liver which acts to obscure operating field, and to mitigate the risk of bleeding or fracture of fibrofatty liver during the traumatic retraction. Most bariatric procedures include the reduction of the volume of the stomach and/or the creation of a small gastric pouch, which entail specific early, late, and life-long nutritional management. The manuscript could benefit from a brief discussion of these clinical aspects of bariatric surgery and a comparison of the role of Mediterranean diet versus other weight loss diets (e.g., low calorie diet, low carbohydrate, low fat diets) for these purposes.

Second, although not necessarily focused on bariatric patients, empirical evidence showing the effectiveness of adherence to MD on weight loss and cardiovascular health are abundant. The authors may want to elaborate the added value of summarizing the evidence in patients with morbid obesity, and how initial BMI may modify the association between MD adherence and weight loss and other clinical outcomes.

Third, it would be helpful to comment on the methodologic differences in measurements (particularly those related to MD adherence), sample characteristics, and analytic strategy.  

Fourth, the included studies were all based on small clinical (convenient) samples. The study that examined preoperative MD adherence used data from only 37 male patients. External validity is an obvious limitation which should be discussed. Moreover, weight loss and comorbidity evolution are multifactorial in nature. Many other factors can influence these outcomes (e.g., physical activity, initial BMI, socioeconomic status, psychosocial and cognitive factors, and social support). These observational studies clearly could not rule out these confounders. The overall conclusion that “Overall, the evidence is clear. Adherence to a Mediterranean diet is directly correlated with weight loss before or after bariatric surgery” represents an over-stretch of the available evidence summarized in the manuscript.      

Reviewer 3 Report

This is a very interesting and well described study. The topic is very important in obesity management.  

Little changes are needed. The references must be updated and I suggest the following papers. In the introduction a stronger advice to bariatric surgery is needed. I would add this work, comparing modern surgical techniques. A brief mention to the importance of follow-up is also needed and a citation is needed. A brief evaluation of the psychiatric component of diet and bariatric surgery is also needed and citations are provided.

Suggested References

 1)

Updates Surg

. 2017 Mar;69(1):101-107.

doi: 10.1007/s13304-017-0426-z. Epub 2017 Mar 6.

Long-term effects of laparoscopic sleeve gastrectomy versus Roux-en-Y gastric bypass for the treatment of morbid obesity: a monocentric prospective study with minimum follow-up of 5 years

Federico Perrone 1, Emanuela Bianciardi 2, Simona Ippoliti 3, Jennifer Nardella 3, Francesco Fabi 3, Paolo Gentileschi 4 5

2)

Bariatric Surgical Practice and Patient CareVolume 10, Issue 3, Pages 119 - 1251 September 2015

Frequent Follow-Up Visits Reduce Weight Regain in Long-Term Management after Bariatric Surgery

Lombardo M.a, bSend mail to Lombardo M.,Bellia A.b,Mattiuzzo F.c,Franchi A.a,Ferri C.a,Padua E.a, b,Guglielmi V.b,D'Adamo M.b,Annino G.e,Gentileschi P.d,Iellamo F.b, e,Lauro D.b

Round 2

Reviewer 1 Report

In this revised manuscript, the author provided the process of paper selection and also some statistic analysis.
The content of the revised manuscript is also clearer and more complete.
I would like to recommend the publication of this revised manuscript

Reviewer 2 Report

The authors have addressed my concerns. I have no additional comments.